# Dissolution Kinetics of Meloxicam Formulations Co-Milled with Sodium Lauryl Sulfate

**DOI:** 10.3390/pharmaceutics14102173

**Published:** 2022-10-12

**Authors:** Jan Patera, Pavla Němečková, Petr Zámostný

**Affiliations:** Department of Organic Technology, University of Chemistry and Technology Prague, Technická 5, 166 28 Prague, Czech Republic

**Keywords:** meloxicam, sodium lauryl sulfate, co-milling, dissolution rate, enhanced dissolution, dissolution kinetics, surface modification

## Abstract

Meloxicam (MLX) is a poorly soluble drug exhibiting strong hydrophobicity. This combination of properties makes dissolution enhancement by particle size reduction ineffective; therefore, combined formulation approaches are required. Various approaches were investigated in this study, including milling, solid dispersions, and self-emulsified lipid formulations. Whereas milling studies of MLX and its co-milling with various polymers have been reported in recent literature, this study is focused on investigating the dissolution kinetics of particulate formulations obtained by co-milling MLX with sodium lauryl sulfate (SLS) in a planetary ball mill with 5–25 wt.% SLS content. The effects of milling time and milling ball size were also investigated. No significant reduction in drug crystallinity was observed under the investigated milling conditions according to XRD data. For the dissolution study, we used an open-loop USP4 dissolution apparatus, and recorded dissolution profiles were fitted according to the Weibull model. The Weibull parameters and a novel criterion—surface utilization factor—were used to evaluate and discuss the drug release from the perspective of drug particle surface changes throughout the dissolution process. The most effective co-milling results were achieved using smaller balls (2 mm), with a co-milling time of up to 15 min SLS content of up to 15 wt.% to increase the dissolution rate by approximately 100 times relative to the physical mixture reference. The results suggest that for hydrophobic drugs, particle performance during dissolution is very sensitive to surface properties and not only to particle size. Co-milling with SLS prepares the surface for faster drug release than that achieved with direct mixing.

## 1. Introduction

Poorly soluble drugs (classes II and IV of the biopharmaceutical classification system (BCS)) represent an important segment of marketed drugs. Improving drug solubility and release kinetics is therefore a continuing challenge that can be approached in many ways on several levels. The milling, nano-milling, and/or co-milling process for drugs with optional additional excipients provides a means to change the phase composition of the drug [1], execute drug amorphization [2], increase the specific surface area of drug particles, modify the surface [3], and form composite particles. Some drug formulations involving milling demonstrate an increased dissolution rate and bioavailability. For example, griseofulvin was one of the first drugs to have its bioavailability enhanced by milling; other examples include nifedipine, ibuprofen, and spironolactone [4].

Milling of pure drugs is subject to certain efficiency limits. Reducing the particle size to a submicron level increases the specific surface area of the drug but is also associated with an increased surface energy of such particles, facilitating particle agglomeration. This agglomeration effectively reduces the drug-free surface area available for dissolution. In a water-based dissolution environment, the drug hydrophobicity contributes to agglomeration, so simple milling or micronizing may not represent a viable choice to increase the drug dissolution rate for hydrophobic drugs [5]. For extremely hydrophobic drugs, co-milling can alter the surface energy and therefore the surface wettability. Because surfactants, such as sodium lauryl sulfate (SLS), are often used to improve the drug particle wettability [6], common application and co-milling can be expected to exert a synergic effect, enhancing the drug wettability, as well as the dissolution rate, which may exceed the combination effect observed for the physical mixture reported by Slamova et al. [7].

Meloxicam (MLX) was used as a model substance in this study. It is known as a sparingly soluble substance in water, belonging to the BCS II class, so its dissolution is the rate-controlling step of the drug pharmacokinetics [8]. Meloxicam solubility is reported at 4 mg/mL saturation concentration in water, and it is extremely hydrophobic and poorly wettable. Belonging to the non-steroidal anti-inflammatory drug (NSAID) group, it is used for the treatment of joint diseases, including rheumatoid arthritis, osteoarthritis, Bechterew disease, etc.; its enhanced bioavailability can both improve drug performance and offer new prospects in terms of therapeutic applications. Milling was previously reported as an option to improve the dissolution of other drugs in the same pharmacological group, specifically piroxicam [9], naproxen, and indomethacin [10]. However, milling was found to be inefficient due to poor drug wettability, so advanced formulation approaches were also reported, for example, by Samprasit et al., using the cyclodextrin complexes [11].

Meloxicam has been the subject of several studies focused on improving its dissolution rate by co-milling, seeking the advantage of the combined effect of particle size reduction and effective particle separation. Kürti et al. [12] investigated the co-milling of MLX in a high-energy planetary ball mill with polyvinylpyrrolidone (PVP) and polyethylene glycol (PEG) polymers. A similar study was published by Agusti et al. [13], who used polymers with small particle sized as stabilizers in the milled product, which was associated with a change in MLX crystallinity during the milling process. Hydroxypropylmethylcellulose (HPMC) polymer was also found to facilitate the MLX phase transition to an amorphous state in a study by Zaini et al. [14]. Amorphization occurred with a wide range of formulation ratios of MLX:HPMC, from 1:1 to 1:5. Etman et al. [15] reported an increased dissolution rate of MLX in tablets prepared using co-milled formulations of MLX and PEG 6000 in a 1:4 ratio. Bartos et al. [16] performed a study on a wide range of milling and co-milling formulations of MLX using wet processing, examining the effects of the presence of a polyvinyl alcohol (PVA) co-former and ball and bead mill process parameters on the obtained particle size. The authors reported amorphization of MLX in the presence of PVA, similarly to the abovementioned studies, which used different polymers. Additionally, co-milling led to a reduction in the obtained particle size (D_50_ = 2.96 μm) compared to milling (D_50_ = 3.55 μm) after 10 min, enabled the production of a nano-sized product (after 50 min, D_50_ = 126 nm). In contrast, under mild co-milling conditions, amorphization did not occur in mixtures with chitosan [17]. The poor wettability of MLX was addressed by Marinko et al. [18] as the crucial factor affecting the performance of the milled formulation.

Despite the relatively intensive research on MLX co-milling to provide formulations with improved dissolution rates and known wettability problems, no studies have been dedicated to the co-milling of MLX with SLS co-former. Therefore, the aim of the present study is to examine the possible MLX dissolution enhancement resulting from co-milling with SLS and to explore the relationship between the structure of mixed and co-milled formulations and their dissolution performance. In this broader context, only a few studies have been published involving SLS: that of Dehghan et al. [19] and that of Aejaz et al. [20], in which SLS was used as the third substance to form ternary solid dispersions comprising PEG 6000 or PVP. In these studies, the addition of SLS exhibited the fastest dissolution, supporting the objective with respect to its selection as the co-former. SLS was chosen as a promising co-milling candidate based on the results of previously reported studies, for example, for gliclazide and nateglinide, which are also BCS class II drugs [21,22].

## 2. Materials and Methods

### 2.1. Materials

Meloxicam (MLX) was obtained from Zentiva Group, a.s. (Prague, Czech Republic). Sodium lauryl sulfate (SLS) was purchased from Fluka (Buchs, Germany). Tween^®^ 80 was purchased from Sigma Aldrich (Prague, Czech Republic). Potassium dihydrogen phosphate and sodium hydroxide were purchased from Penta a.s. (Prague, Czech Republic).

### 2.2. Particle Size Analysis

Particle size was measured measurement and the particle size distribution (PSD) was determined using a static light-scattering method on a Mastersizer 3000 equipped with a Hydro MV wet dispersion unit (Malvern Instruments Ltd., Malvern, UK). Demineralized water was used as the liquid dispersant. A combination of red and blue lasers was used to measure the particle size. The following fixed parameters were used: refractive index of liquid medium and measured substance: 1.33 and 1.72, respectively; bulk density of MLX and medium: 1.613 g/cm^3^ and 1 g/cm^3^, respectively; and absorption coefficient of MLX: 0.1. The optimized measurement parameters were set as follows: stirrer rotation speed of the Hydro MV dispersion unit: 50%; measurements per sample: 5; and sonification: 20%. An adequate amount of sample was added to the dispersion medium in the wet unit with a set stirring speed or sonification with the addition of drops of surfactant (Tween 80) to the active substance due to its low wettability, leading to agglomeration of particles. The suspension was stirred in the dispersing unit for 5 min; then, an additional amount of MLX was added to increase the obscuration to 10–15%. Three series of measurements were performed with an interval of 5 min.

### 2.3. Micronization and Co-Milling of Meloxicam

A PM100 CM planetary ball mill (Retsch GmbH, Germany) was used to prepare reference micronized (R) and co-milled (CM) MLX samples. Reference samples were prepared by weighing one gram of MLX into a 50 mL grinding jar equipped with 10 mm diameter grinding balls. Micronizing experiments were carried out with varying milling times of 5, 10, and 15 min and a rotational speed setting of 300 rpm. At the end of each grinding interval, a sample was taken from the grinding jar, the particle size distribution was measured, and the specific surface area value was recorded; approximately 10 mg of the sample was taken to perform a dissolution experiment on micronized MLX using the flow-through dissolution method (USP4). Further milling experiments were carried out with grinding balls of two different sizes (2 and 10 mm in diameter) at the same rotational speed (100 rpm) for a total grinding time of 25 min (for 10 mm balls) or 30 min (for 2 mm balls). The milling was interrupted at 5 min intervals to withdraw samples. The particle size distribution was measured, and the specific surface and Sauter mean diameter (surface/volume moment mean, *D* [3,2]) values were determined for each sample.

Co-milling experiments of MLX with SLS were carried out using both 2 mm and 10 mm grinding balls with SLS mixture content of 5, 15, and 25 wt.% and a total sample mass of 1 g. The rotation speed of the mill was set to 100 rpm, and the grinding time ranged from 5 to 35 min. To determine the influence of SLS alone (without co-milling) on the dissolution behavior, micronized reference MLX was mixed in a Turbula^®^ T2F 3D mixer (Willy A. Bachofen AG Maschinenfabrik, Basel, Switzerland) with 15 wt.% SLS at 100 rpm for 10 min and tested as a control (reference) sample.

Table 1 summarizes all the studied samples, which are labeled according to the type of preparation (reference (R) or co-milled (CM)). The first index represents the size of grinding balls (10 mm or 2 mm), the second index represents SLS content (in wt.%), and the third index indicates (co-)milling time (in minutes).

### 2.4. Dissolution Testing

The release rate of MLX from reference and co-milled samples was studied using the flow-through dissolution method. Dissolution studies were carried out in the USP 4-compliant flow-through cell apparatus Sotax CE1 (Sotax, Basel, Switzerland) with a Sotax CY1 piston pump (Sotax, Basel, Switzerland). A dissolution flow-through cell for powders and granules with a diameter of 12 mm and a height of 32 mm was used to study the samples in all experiments. Each experiment was conducted using the cell in an open-loop system with a fresh dissolution medium from the reservoir continuously passing through the cell. An open-loop system was selected due to the low solubility of MLX and the requirement for a high volume of solvent. A dissolution medium with a pH of 7.2 containing 6.8 g of potassium dihydrogen phosphate and 0.9 g of sodium hydroxide dissolved in 1000 mL of demineralized water was degassed in an ultrasonic bath for 15 min prior to measurement. The test sample was finely spread with a pestle in an agate mortar and weighed directly into the flow cell. All samples weighed between 10 and 13.5 mg to maintain approximately the same amount of MLX, even in mixtures that contained SLS. The dissolution medium and the cell were placed into a water bath and heated to 37 °C. The flow rate of the dissolution medium through the cell was set to 23 mL/min. Samples were collected into beakers at intervals ranging from 0 to 30 min.

### 2.5. UV-VIS Spectroscopy

The concentration of MLX in samples taken during dissolution tests was determined by UV-VIS spectrophotometry using a Specord 200 Plus spectrophotometer (Analyst Jena AG, Jena, Germany). Graphs of the effect of MLX calibration solution concentration on absorbance were plotted based on the measured absorbance of the calibration solutions at two wavelengths (271 nm and 363 nm) for high- and low-concentration samples. MLX concentrations in the samples were calculated using equations obtained by linear regression.

### 2.6. SEM Analysis

The morphology of powder and the distribution of individual elements were observed with a TESCAN LYRA3-GMU scanning electron microscope (Tescan, Brno, Czech Republic) at an acceleration voltage of 10 kV. Elemental analysis was performed using an energy-dispersive spectroscopy (EDS) analyzer with a 20 mm^2^ X-maxN EDS SDD detector (Oxford Instruments, Oxford, UK) in back-scattered secondary electron (BSE) imaging mode. A small amount of mixture was placed on a copper adhesive conductive tape and coated with a 10 nm gold layer to ensure the electron conductivity of the sample.

### 2.7. XRD Analysis

X-ray powder diffraction data were collected at room temperature with an X’Pert3 Powder θ-θ powder diffractometer with parafocusing Bragg–Brentano geometry using CuKα radiation (λ = 1.5418 Å, U = 40 kV, I = 30 mA). Data were scanned with a PIXCEL ultrafast 1D detector in the angular range of 5–70° (2θ) with a step size of 0.039° (2θ) and a counting time of 115.26 s step^−1^. Data were evaluated using HighScore Plus 4.0 software.

## 3. Results and Discussion

The hydrophobic drug meloxicam was co-milled with an SLS co-former in a planetary ball mill to increase its dissolution rate. The effect of co-milling on the dissolution properties of MLX was evaluated by dissolution tests. SLS was selected as a suitable co-former for co-milling with MLX based on the results of previously published papers [18,19,20,21]. Furthermore, the parametric sensitivity of the co-milling process to the choice of milling ball diameter, grinding time, and SLS content was studied in terms of the degree of influence on the dissolution properties. We used mild milling conditions to prevent undesired amorphization of the drug. SLS was chosen such that even the highest value (25 wt.%) corresponded to a maximum of 1.5 wt.% in the final dosage form.

### 3.1. Meloxicam Particle Size Development during Milling

Drug particle size is a major factor affecting the dissolution process. Because particle size effects could interfere with those related to particle structure obtained by co-milling, we aimed to closely monitor particle size during the experiments to determine possible changes in the drug specific surface in dependence on the grinding time, ball size, and mill speed. MLX particle size distribution (PSD) curves were obtained using the static light-scattering method at individual milling time intervals for grinding balls of 2 and 10 mm in diameter and mill rotational speeds of 100 and 300 rpm, respectively. Figure 1a shows the development of MLX PSD milled at 300 rpm using 10 mm balls at various time intervals. The starting pure MLX (R0) exhibits bimodal PSD with modes at approximately 1 μm and 10 μm. The proportions of the largest particle sizes of MLX gradually decrease with increased grinding time, and the fine fraction increases. Simultaneously, the distribution modes shift toward lower values. In contrast, under the condition of grinding at 100 rpm for 15 min (Figure 1b), the volume fraction of fine particles is reduced by approximately half. Furthermore, Figure 1a shows that the distribution varies from bimodal with peaks around 8 μm (primary MLX particles) and 1 μm (dust fraction) to almost monomodal with a continuously increasing fraction of particles 0.5 μm in diameter and larger.

Figure 1b shows that the maxima of the MLX distribution curves under the condition of 100 rpm and 10 mm balls are shifted toward lower particle size values (approximately 5.5 µm) compared to those milled using 2 mm grinding balls at the same speed due to the greater energy applied to the material as a result of the impact of the bulky balls on the MLX particles during the grinding process. The PSD curve for a sample milled for 15 min using 2 mm balls at 100 rpm is almost identical to that obtained in the experiment using larger milling balls for 10 min at a rotation speed of 300 rpm. Differences in measured values between pairs were assessed on the basis of a *t*-test of mean values of Sauter diameter and standard deviations calculated from 10 replicates. Rotation speed does not have a significant influence at the 0.05 level (*p*-values range from 0.24 to 0.88). Using a higher rotation speed provides results with a similar mean particle size but with a larger standard deviation corresponding to occasional reaggregation of particles. Based on these results, a 100 rpm rotation speed was used for all co-milling experiments to suppress the comminution effect compared to surface modification, which is the objective of the present study.

The MLX specific surface area values following the investigated milling treatments were estimated in terms of PSD as a quantitative descriptor of the comminution effect. PSD was evaluated with Mastersizer 3000 software, assuming smooth, non-spherical particles, in accordance with the particle morphology discussed later in this article. The obtained specific surface area data, together with Sauter diameter (Table 2), were subsequently used to evaluate the MLX release rate, specifically to calculate the surface utilization factor as described below.

### 3.2. Comparison of Meloxicam Release from Different Formulations Using USP4

Dissolution tests were performed to evaluate the release rate of the reference MLX with characteristic PSD and that of MLX from SLS-containing samples. MLX release rates were compared in terms of preparation method (co-milling and simple mixing) and SLS:MLX ratio. Furthermore, we evaluated the effect of the size of the grinding balls and the time of co-milling on the dissolution properties of MLX.

The individual profiles of MLX release were compared using the dissolution profiles of the relative rate of MLX release, the cumulative amount of MLX released, and the surface utilization factor over time.

Dissolution tests were performed in an open-loop flow cell as described in the experimental section. The same parameters were set for all tested mixtures, and the same dissolution medium was used. The measured absorbances of the samples were recalculated according to MLX concentrations. At least two experiments were performed for reference samples without SLS, and at least three experiments were performed for co-milled samples with a similar weight to verify the reproducibility of the results. The results of the dissolution tests are presented as means of the values with standard deviations (for n ≥ 3 experiments).

#### 3.2.1. Screening of Dissolution Kinetics According to the Weibull Model

To obtain an initial overview of the effect of SLS co-milling, dissolution experiments were performed using the co-milled formulations containing varying levels of SLS. The dissolution profiles were recorded (Figure 2). Among the tested samples to optimize the content of the excipient, the best-performing mixtures were CM10-25/10 (25 wt.% SLS) and CM10-15/10 (15 wt.% SLS). SLS addition of between 15 and 25 wt.% is most the promising with respect to enhancement of the dissolution. For further analysis, the data were fitted with an empirical Weibull model, the parameters of which enable a rough quantitative comparison of the individual formulations. The Weibull model is a mathematical model that is often used to describe the kinetics of drug release from a dosage form. The general empirical dissolution equation described by Weibull was adapted to the dissolution process. When applied to the release of drug from a pharmaceutical dosage form, Equation (1) [23] expresses the amount of drug released into the solution:*W*(*t*) = *W*_0_(1 − exp(−*k*·(*t* − *T*_0_)^b^)),(1)
where *W(t)* is the accumulated amount of dissolved drug dependent on time *t*. The amount can be expressed in units of weight or relatively, in the form of a weight fraction. In this work, the dissolution profiles are expressed in accordance with pharmaceutical practice in the form of a weight fraction of the drug released into the solution relative to the total amount of loaded drug. The parameter *T*_0_ represents the initiation period or characteristic time delay associated with the disintegration of the dosage form. The parameter *k* is the dissolution rate constant, which is sometimes expressed in Equation (1) as *k =* 1*/a*, where the *a* describes the measure of the dependence of *W(t)* on time. Parameter *b* is referred to as the shape parameter and describes the shape of the dissolution curve [24].

With respect to the value of parameter *b*, three cases are generally observed [24,25]:For *b =* 1, the shape of the curve exactly corresponds to the shape of the exponential function described by Equation (1). The dissolution rate follows the first-order kinetics (i.e., the dissolution rate is dependent on the residual amount of drug in the dosage form). The dissolution rate is not constant but decreases as the amount of drug in the dosage form decreases.For *b <* 1, the curve has an elongated shape in the direction of the timeline, i.e., less steep compared to the shape of the curve in case 1). Dissolution proceeds more slowly than according to first-order kinetics. A hindering effect is observed, which often occurs in association with sustained release dosage forms, which may be caused by the diffusion barrier of the gel layer formed by the hydrophilic polymer matrix or the gradual change in the rough surface of the particles to a smooth surface during dissolution.For *b >* 1, the shape of the curve is sigmoid with one inflection point. The dissolution rate does not change monotonously but increases non-linearly up to the inflection point and then decreases asymptotically.

The mathematical model was used to perform non-linear regression analysis using ERA [26] software. Parameters *k*, *b*, and *T*_0_ were estimated using the least-squares objective function. Estimated parameter values for all tested MLX samples are summarized in Table 3 and Table 4.

In accordance with published data [21,27], our experiments confirmed that the SLS content in the mixture is critical to the optimal size of the SLS particle area available to effect MLX particle dispersion throughout the dissolution medium volume and to promote drug–solvent interaction.

The CM10-15/10 mixture showed the most uniform profile of dissolution, with a limited tendency to slow down with increased time, reflecting the high value of parameter *b* (Table 3). Differences in calculated values of the Weibull model between pairs were assessed on the basis of a *t*-test of mean values and standard deviations calculated from three replicates. For parameter *k*, differences were insignificant at the 0.05 significance level (with *p*-values 0.09 and 0.68). The *b* value of CM10-15/10 was significantly higher (*p*-values 0.027 and 0.0005); therefore, 15 wt.% SLS content was chosen for further experiments.

Given the very low to negligible *T*_0_ values in the range of ~0–0.4 min, the disintegration or dispersion of the dosage form is not significant in dissolution experiments, confirming that the experimental results were not influenced by the formation of any disintegrable powder agglomerates and that the course of dissolution corresponds to the powder form.

The reference MLX samples exhibited more than 50× slower release relative to that of co-milled samples. Within the samples without added SLS, the dissolution kinetics profiles were dependent on milling time, as the starting sample (R0) showed the smallest *b* value, which increased to nearly to 1 after micronization for 5 min (R10-0/5), reaching a maximum *b* value as non-isometrical particles were broken. Further milling increased the particle angularity and surface area, with R10-0/10 and R10-0/15 samples exhibiting a gradual decrease in the *b* value and an increase in the *k* value. This reflects the process of gradual change of anisometric particles with a rough surface to more isometric particles with a smooth surface via surface dissolution, with the particle surface/volume ratio decreasing and the dissolution slowing down more than it would in correspondence to first-order kinetics. The value of the dissolution rate constant (*k*) for sample R10-0/5 insignificantly decreased (*p*-value 0.35) compared to the value of the untreated sample of pure MLX with characteristic PSD (R0). However, a significant increase in *b* (*p*-value 0.003) was also observed, indicating that the rate of dissolution did not decrease substantially but changed its kinetic progression.

A comparison of the dissolution profiles (Figure 3) of the reference micronized sample of pure MLX (R10-0/10) with those of mixed samples shows that mixing with SLS (R10-15/10) leads to a large increase in the dissolved amount of MLX over a 30 min monitored period. Nevertheless, co-milling leads to a further substantial dissolution enhancement, which depends on the co-milling time, independent of the size of the grinding balls. Due to the higher energy impact observed with 10 mm balls, the dissolution exhibits steeper profiles for the shorter co-milling times. With the increased length of co-milling, the release rates of MLX from the mixtures diminish. A comparison of the values of release rate parameters obtained with Weibull model is shown in Table 4.

The values of the constant rate (*k*) indicate that the milling or mixing of MLX with SLS excipient significantly increases the dissolution rate. All *k* values for MLX samples prepared by co-milling (CM-) are greater than those for the mixture prepared by blending micronized MLX with SLS (R10-15/10 in Table 3), with *p*-values based on the *t*-test range of 0.001–0.047, demonstrating that co-milling is effective in terms of improving the dissolution properties of MLX through various mechanisms. During the process of co-milling MLX with SLS, the following phenomena were observed: an ordered mixture in which the MLX is in close contact with the SLS [28], hydrophilization of the MLX surface by increasing the polar surface energy component of the MLX particles because they can interact with weak secondary bonds [29], an increase the wettability of MLX particles by SLS [21], an increase the specific surface area of the particles, and an increase in the angularity of the particles [4]. The crystallinity of the investigated samples did not differ significantly, as confirmed by XRD measurements (Figure 4a); a decrease in the intensity (count) of diffracted signals of MLX was not visible. When comparing the diffraction patterns of pure MLX (R0) with the co-milled samples, no visible changes in the intensities or width of the characteristic diffraction lines, nor the elevated background, are evident that would characterize the amorphization of the sample to a large extent. Compared to the results reported in previous studies [14,15,16], we observed no quantitative amorphization of meloxicam. Therefore, amorphization is not an influencing parameter for such changes in the dissolution behavior of MLX. Given that the bulk is crystalline and the dissolution was monitored in a wide range, there was no need to characterize trace amorphization by other techniques. The only differences between the diffraction patterns shown in Figure 4 are intensities of peaks at 7° and 21° of 2-theta, which correspond to the SLS and change only with its content in the mixtures; the sample with the highest SLS content (CM10-25/10) shows the highest intensity of these peaks (Figure 4b).

The highest overall value for the parameter *k* was observed for the sample CM2-15/30, which was prepared by co-milling for 30 min at 100 rpm using 0.2 cm grinding balls with 15 wt.% SLS. The value of parameter *b* was less than one for all test samples, in accordance with the above-described case wherein dissolution proceeded slower than would correspond to first-order kinetics; however, *b* values were higher for co-milled samples compared to the mixed reference, proving the efficiency of SLS co-milling in terms of improving dissolution over the whole time interval. This result is graphically illustrated in Figure 3, where the initial slopes of all CM dissolution profiles are only slightly or moderately better than that of the reference mixed R10-15/10 sample, releasing 80% of the drug between 10 and 25 min of the dissolution, whereas the reference mixed sample released only 70% of the drug up to 30 min. The best-performing CM samples released more than 95% of the drug within 15 min. Compared to the results reported in previous studies, SLS appears to be the most suitable candidate to improve the dissolution characteristics of MLX by the co-milling technique. Unlike HPMC or chitosan, which achieve similar results in dissolution tests (release of up to 90 wt.% of MLX in 15 min), SLS can be used at a much lower concentration. Chitosan achieves similar results to MLX in a ratio of up to 8:1 [17] and HPMC in a ratio of 3:1 [14]. Other co-formers, such as PVP [12,13], PEG 6000 [15], and PVA [16] achieved slightly worse results in dissolution tests, for which the dissolved amount of MLX reached a maximum of 70% in 30 min.

#### 3.2.2. Analysis of the Relative Release Rate of Meloxicam

Because the co-milled samples provided performance enhancement in a sustainable manner, enabling the release of a much higher fraction of the drug in a short time, it is interesting to investigate the dissolution process in detail, exploring the release rate development over time for different types of samples. The relative release rate is defined as the MLX mass flow from the dissolution cell relative to the mass of MLX in the test sample. Assuming ideal cell mixing, it is also equal to the relative mass flow of MLX in the liquid phase. The relative release rate was calculated according to Equation (2):*r* = (*c*_MLX_·*Q*)/*m*_MLX_,(2)
where *r* denotes the relative release rate, *c*_MLX_ is the immediate MLX concentration in the sample, *m*_MLX_ is the initial weight of MLX in the test sample, and *Q* is the volumetric flow rate of the dissolution medium [7].

A comparison the relative release rate profiles (Figure 5) and graphs of MLX release from the prepared mixture by simple mixing or milling (at the same SLS:MLX ratio) over time shows that co-milling, independent of the set process parameters, guarantees dissolution of a higher amount of the MLX than with the mixing method (Figure 3), which also corresponds to the dissolution rate constant of these mixtures calculated according to the Weibull model (Table 3 and Table 4). The continuous impact of the grinding balls on the particles more effectively breaks down the agglomerates of hydrophobic MLX particles, thereby increasing their wettability.

During the mixing of micronized MLX with SLS, MLX aggregates for, reducing the wettability of the MLX particles and preventing the solvent from penetrating the surface of the SLS particles. Several authors have indicated that agglomeration can affect the particle size distribution and physical properties and negatively influence the dissolution properties [30,31].

#### 3.2.3. Analysis of the Dissolution Course by the Surface Utilization Factor

Because the surface of particles available for dissolution and the processes taking place on the composite surface are crucial for understanding the dissolution behavior, we introduced the surface utilization factor, *η = A_ef_/A_b_*, as a dimensionless ratio of effective surface area (*A_ef_*) and balance surface area (*A_b_*).

The effective surface area (*A_ef_*) expresses the surface factor of the Noyes–Whitney Equation (3) [32].
(3)$$\dot{m}\left(t\right)=k\cdot A_{\mathrm{ef}}\left(t\right)\cdot \left(c_{s}-c\left(t\right)\right),$$
where the dissolution rate is expressed by the mass flow of the drug from the dissolution cell ($\dot{m}$), parameter *k* is the dissolution rate constant, *c* is the instantaneous concentration, and *c_s_* is the saturation concentration of MLX in the dissolution medium (experimentally determined as 233.6 mg/L). The dissolution rate constant was calculated using the surface area of MLX particles (Table 2) obtained from particle size distribution measurement using a Mastersizer 3000 at the beginning of the experiment and $\dot{m}$ and *c* values measured at t = 0. Therefore, at t = 0, *A**_ef_* is equal to the initial surface area of the particles in the sample.

The balance surface area (*A_b_*) expresses the surface area of an undissolved portion of MLX particles present in the dissolution cell calculated using Equation (4) and assuming a spherical particle shape.
(4)$$A_{b}\left(t\right)=N\cdot \pi \cdot d^{2}=\frac{m_{\mathrm{MLX}}\left(0\right)}{\rho \cdot \frac{\pi }{6}\cdot d{\left(0\right)}^{3}}\cdot \pi \cdot {\left[d\left(0\right)-\sqrt[{3}]{\frac{6\cdot \left(m_{\mathrm{MLX}}\left(0\right)-m_{\mathrm{MLX}}\left(t\right)\right)}{N\cdot \rho \cdot \pi }}\right]}^{2},$$
where *N* is the number of particles in the sample cell, *ρ* is the MLX true density of 1.613 g/cm^3^ (calculated using Advanced Chemistry Development (ACD/Labs) software V11.02 (©1994–2020 ACD/Labs)), and *d* is the Sauter diameter of the MLX particles, where the 0 or *t* in parentheses indicates the initial or instantaneous value, respectively. The residual mass of non-dissolved MLX is calculated according to the released *L*(*t*) fraction (5).
(5)$$m_{\mathrm{MLX}}\left(t\right)=\left(1-L\left(t\right)\right)\cdot m_{\mathrm{MLX}}\left(0\right).\;$$

According to the above definitions, the surface utilization factor (*η*) is equal to 1 at the beginning of each dissolution experiment and develops throughout the experiment, indicating whether the dissolution surface availability of the drug increases or decreases. The following options and their interpretations are available:η = 1 indicates that the particles are approximately isometric with a smooth surface and that the effective and balance surface areas decrease at the same rate.η < 1 indicates that the particles are either non-isometric, rugged, or broadly distributed and that during dissolution, the effective surface area is reduced at a faster rate than the balance surface area, either due to a reduction in the aspect ratio, a reduction in the surface ruggedness, and/or elimination of fine particles of the broad distribution.η > 1 indicates the presence of composite particles of MLX and SLS. Such composite particles dissolve faster than MLX particles due to the presence of SLS, and MLX fragments are released, so the effective surface area increases, even if particles are partially dissolved.

Figure 6 compares the co-milled mixtures with three different SLS contents (5, 15, and 25 wt.%) with 10 mm grinding balls. The CM10-15/10 mixture shows the most uniform profile, which corresponds to the highest value of parameter *b*, according to Table 3. The initial decrease in the η values shown in Figure 6 is the result of the rapid dissolution of the submicron MLX particles presented in the sample, independent of co-milling time and the ratio MLX:SLS. This initial decrease in DE values corresponds to the dissolution profiles shown in Figure 2. In the CM10-15/10 mixture, submicron particles and particles within the micron range are well dispersed on the surface of larger particles (Figure 7c). In contrast to micronized MLX without SLS (Figure 7a), good particle dispersion without agglomeration of the fine particles, as well as a shift of the center of gravity of the particle size distribution to the order of micron units, is observed.

The performance of the co-milled formulation was found to be sensitive to sodium lauryl sulfate, meloxicam ratio, and milling parameters. Increasing the SLS fraction to 15 wt.% facilitates surface activation and stabilization, with those formulations outperforming those with lower SLS content. On the other hand, increasing the SLS fraction leads to the different structures of the particles; the formulations with 25 wt.% SLS content are significantly inferior in terms of the dissolution rate.

The co-milling of SLS and MLX with a high SLS content (CM10-25/10) produces a composite (Figure 7d). The surface of the composite shows a high angularity caused by the presence of SLS particles, resulting in disintegration of the composite into fragments in the dissolution medium. This increases the total available dissolution surface, so the *η* values increases beyond one relative to the initial values, and dissolution is accelerated (Figure 6). The CM10-5/10 mixture (Figure 7b) also shows an increase in the *η* values to the maximum, although lower than the *η* values of the CM10-25/10 mixture, as it is probably composed of at least two types of units, the latter of which did not disintegrate during dissolution.

The composite structure can be visualized using EDS imaging, which is sensitive to the elemental composition of the surface. The sulfur:sodium (S:Na) ratio was used to distinguish image regions corresponding to MLX (does not contain sodium) and SLS regions (containing sodium). The two types of composite units can be seen on the SEM image with the EDS map (Figure 8). Regions with dark red spots with more S:Na atoms are visible, indicating the presence of larger particles or clusters of MLX, characterized by a smooth surface and smaller SLS particles deposited on them (middle-top and bottom). The second type of unit represents clusters with highly angular particles formed by SLS particles with deposited MLX microparticles (bottom-right and middle-left). In Figure 8b, these clusters are represented by green regions of higher concentration of Na atoms covered by a thin layer of MLX atoms represented by red color.

The type of resulting mixture prepared by milling using 2 mm grinding balls is affected by the co-milling time. The mixtures milled for 5 to 15 min shown in Figure 9a show an initial increase in *η* values, which is the highest for the milled mixture for the shortest time, which apparently contains coarser SLS particles on which fine MLX particles are wedged. After reaching the maximum, the *η* values decrease due to the gradual dissolution of SLS particles and the reduction in the fine particle fraction. DE still holds above one for 10 min but falls below after 15 min. The lowest *η* value at 30 min was recorded for the mixture milled for the shortest time (CM2-15/5) because it contained the smallest proportion of submicron particles.

The surface utilization factor of mixtures co-milled for 20 min or longer decreases relative to the baseline and is less than 1 (Figure 9b), suggesting that longer co-milling may result in a composite composed of multiple units that do not disintegrate in the dissolution medium and are dissolved only from the surface. During the first 30 s, the *η* values drop sharply. The greatest decrease is observed in the 35 min mix, possibly as a result of agglomerates of hydrophobic MLX particles reducing the total value of the available dissolution surface. Conversely, a gradual decrease is observed for the CM2-15/30 mixture, which was characterized by the highest dissolution rate constant (*k*) according to Weibull’s evaluation (Table 4). The most stable MLX dissolution curve, with *η* values stabilizing after 1 min of dissolution, was observed for sample CM2-15/25, which corresponds to the highest value of parameter *b* (Table 4).

Co-milling using coarser grinding balls (10 mm diameter) delivers high energy to the milled material. During the co-milling process, the particles are preferably fragmented, but agglomerates are also formed from the hydrophobic MLX particles. The stability and coarseness of the dissolution depend, to some extent, on the secondary treatment of the samples, i.e., how the aggregates can be spread prior to the dissolution and whether the resulting aggregates are disintegrated during co-milling.

The graph in Figure 10 shows that the most stable course of dissolution was associated with the mixture co-milled for 15 min (CM10-15/15), as its *η* values began to stabilize after 45 s of dissolution. The values of parameters *k* and *b* shown in Table 4 confirm that the release of MLX from the 15 min co-milled mixture was the fastest and, together with the 25 min co-milled mixture, the most uniform compared to the other CM10-15 mixtures.

## 4. Conclusions

The results of the present study show that the dissolution rate of meloxicam can be substantially improved by co-milling with sodium lauryl sulfate using a ball mill. The co-milling process was proven essential for improvement of the dissolution rate, as control samples prepared by separate milling of meloxicam under identical conditions and/or by mixing with sodium lauryl sulfate exhibited much slower dissolution than co-milled samples. The excellent performance of co-milled formulations is caused by surface hydrophilization, suppressing the formation of hydrophobic agglomerates, increasing the meloxicam particle surface disorder induced by milling with sodium lauryl sulfate, and exhibiting the stabilizing effect on those changes. These changes in performance were obtained on the level of particle technology, as no observable changes in the solid-state properties were observed by XRD.

The newly introduced surface utilization factor was used to compare the dissolution performance of various formulations throughout the dissolution experiment. The factor was found to successfully reflect the development of the particle performance in terms of dissolution as particles dissolve and their sizes are reduced. Analysis of the dissolution profiles using the surface utilization factor revealed varying performances of the particles prepared by co-milling under different conditions. Co-milling was found to be most effective under more gentle milling conditions using 2 mm balls, which provided more surface activation than 10 mm balls. The optimum co-milling time was determined to be approximately 15 min, as longer milling does not provide further dissolution improvement and is associated with a risk of forming undesirable agglomerates, even under gentle milling conditions.

In general, it can be concluded that co-milling hydrophobic drugs with surface-active agents may lead to substantial improvement of their dissolution rate, which occurs at the level of particle technology rather than mechanochemistry and solid phase transition. Based on the MLX/SLS example, the particle performance is sensitive to the surface properties but not particle size; therefore, processing conditions tailored to the formation of desired particle structure are more suitable that operating the process at the highest possible energy input. Furthermore, the suggested surface utilization factor profiling can prove advantageous to evaluate the formulation dissolution properties in general and may aid in the optimization of the co-milling processes for other therapeutic systems.

## Figures and Tables

**Figure 1 pharmaceutics-14-02173-f001:**
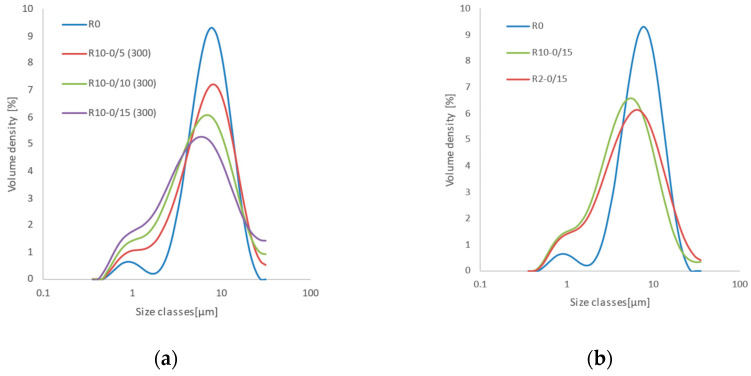
Particle size distribution of reference meloxicam and milled reference samples varied according to the diameter of the grinding balls, grinding time, and rotation speed of the mill: (**a**) 10 mm balls, 5–15 min, 300 rpm; (**b**) 2 mm and 10 mm balls, 15 min, 100 rpm.

**Figure 2 pharmaceutics-14-02173-f002:**
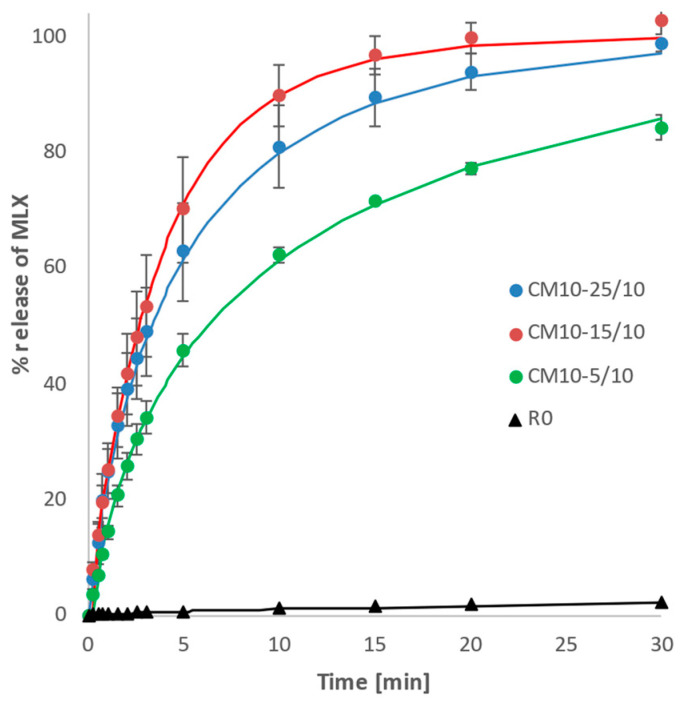
Dependence of MLX released on time for samples co-milled for 10 min with 10 mm balls and with SLS contents of 5, 15, and 25 wt.% and for the reference sample of pure MLX (R0). Experimental points fitted according to the Weibull model.

**Figure 3 pharmaceutics-14-02173-f003:**
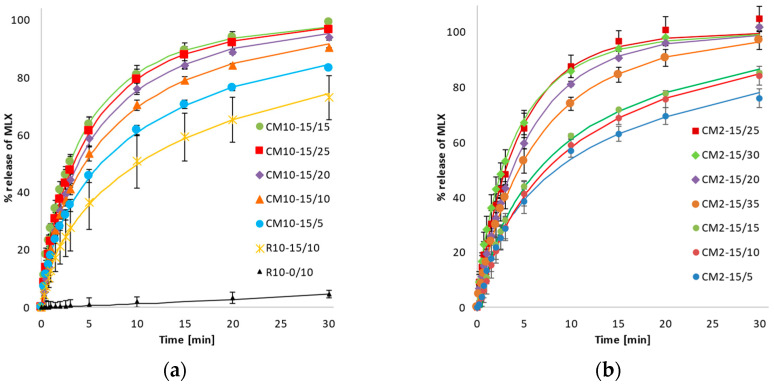
Dependence of MLX released on time for reference and co-milled samples with 15 wt.% SLS content. Experimental points fitted with the Weibull model. Comparison of (**a**) micronized reference samples without (R10-0/10) or mixed with SLS (R10-15/10) and co-milled mixtures with 10 mm balls in the time range of 5 to 25 min; (**b**) co-milled mixtures with 2 mm balls in the time range of 5 to 35 min.

**Figure 4 pharmaceutics-14-02173-f004:**
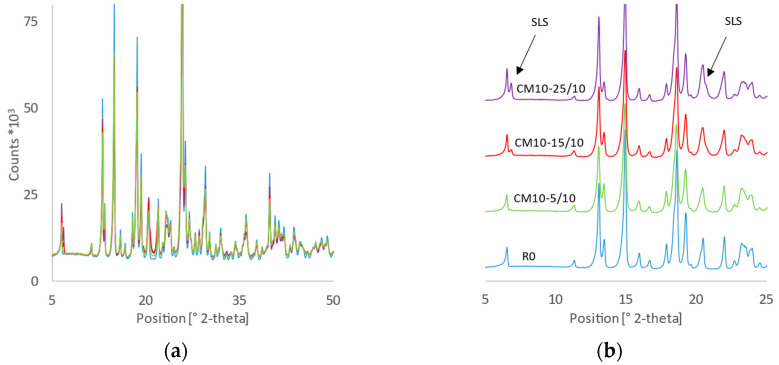
X-ray diffraction patterns (**a**) of reference MLX (R0-blue) and co-milled mixtures with increasing SLS content (CM10-5/10-green; CM10-15/10-red; CM10-25/10-purple); with offset (**b**).

**Figure 5 pharmaceutics-14-02173-f005:**
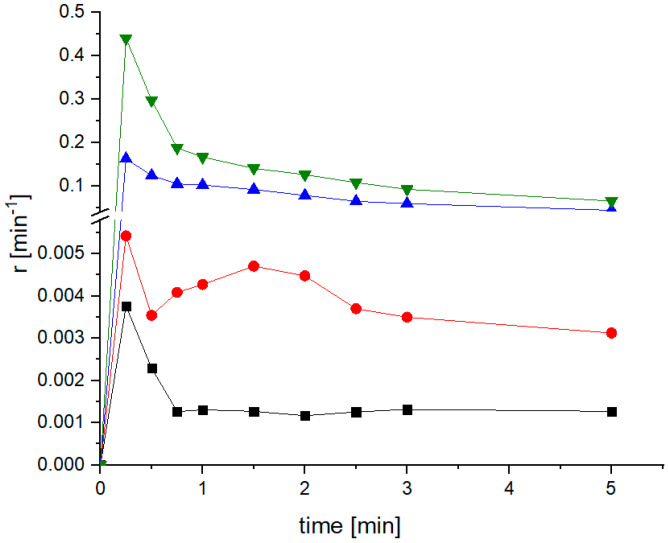
Profiles of the relative release rate of untreated MLX (R0, black) micronized for 10 min (R10-0/10, red); mixed, micronized MLX with 15 wt.% SLS (R10-15/10, blue); and co-milled MLX with 15 wt.% SLS with 10 mm balls for 10 min (CM10-15/10, green).

**Figure 6 pharmaceutics-14-02173-f006:**
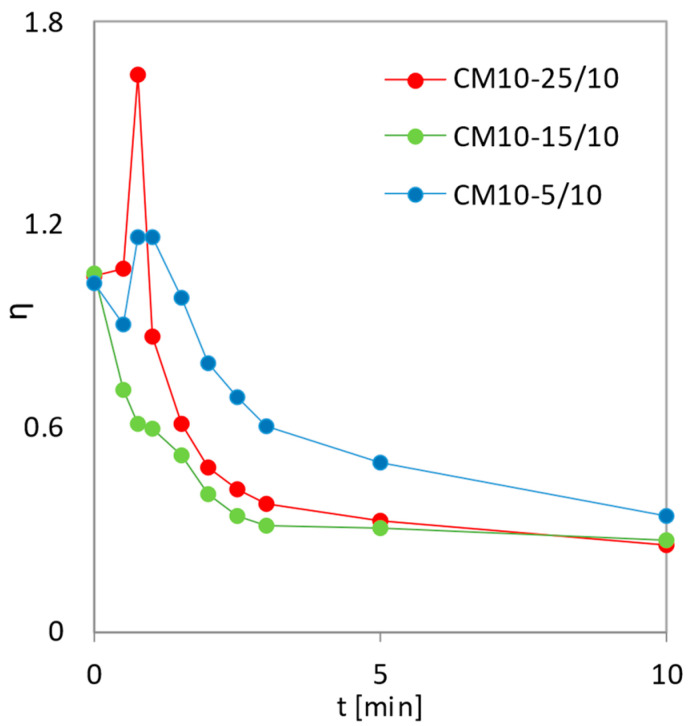
Dependence of the surface utilization factor on dissolution time for co-milled mixtures with varying compositions. Co-milled MLX with 10 mm balls for 10 min; 5, 15, and 25 wt.% SLS.

**Figure 7 pharmaceutics-14-02173-f007:**
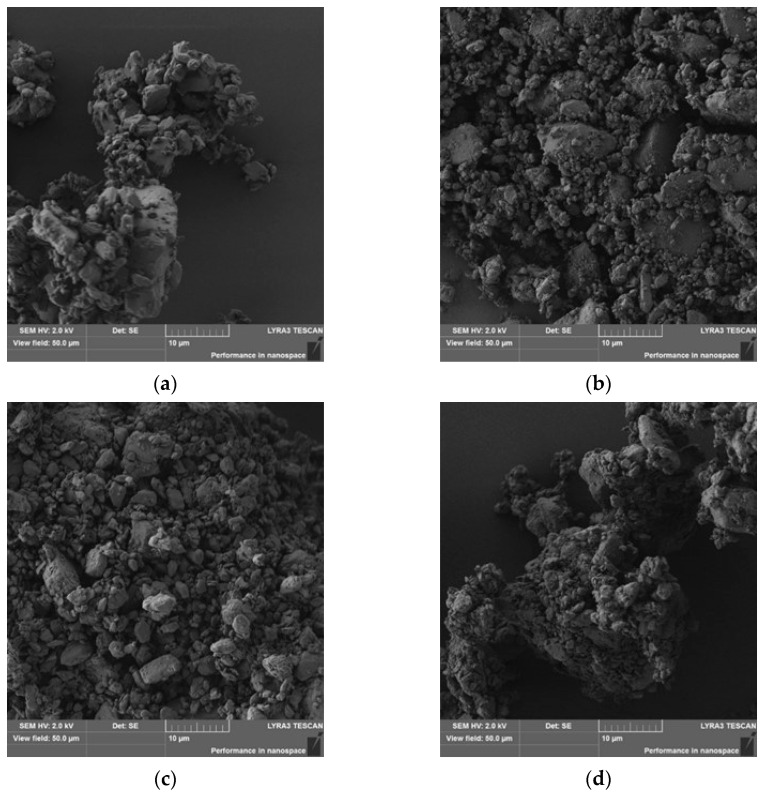
SEM images of the co-milled mixtures of MLX with 10 mm balls for 10 min; (**a**) reference milled sample without SLS (R10-0/10); (**b**) CM10-5/10; (**c**) CM10-15/10; (**d**) CM10-25/10.

**Figure 8 pharmaceutics-14-02173-f008:**
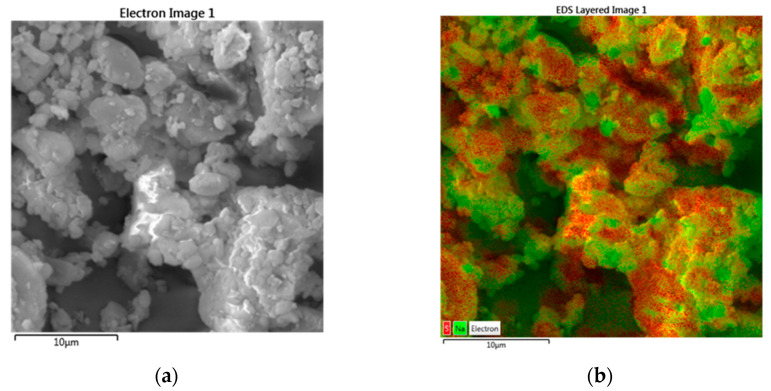
SEM image (**a**) and EDS map (**b**) of the co-milled mixture of MLX with 10 mm balls for 10 min; 15 wt.% SLS (CM10-15/10). Sodium-rich regions are represented as green, and sodium-deficient regions are represented by red.

**Figure 9 pharmaceutics-14-02173-f009:**
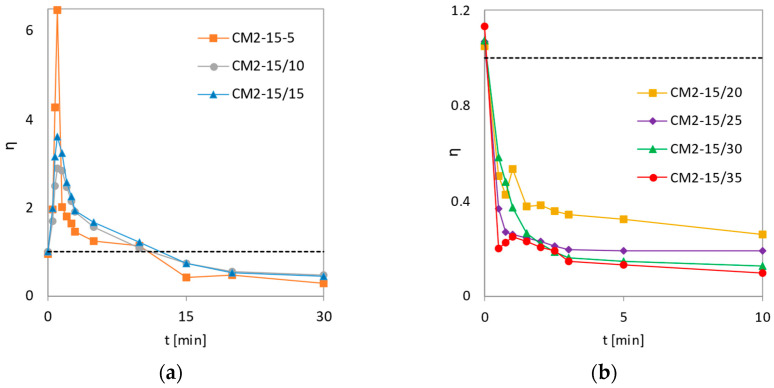
Dependence of the surface utilization factor on dissolution time for co-milled mixtures with 2 mm balls and SLS content of 15 wt.%. The co-milling time was: (**a**) 5, 10, and 15 min; (**b**) 20, 25, 30, and 35 min.

**Figure 10 pharmaceutics-14-02173-f010:**
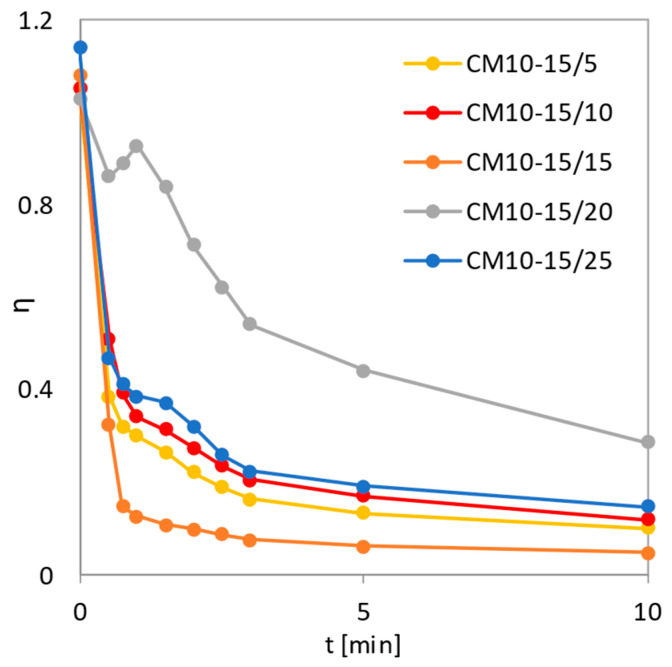
Dependence of the surface utilization factor on the dissolution time for co-milled mixtures with 10 mm balls with 15 wt.% SLS content. The co-milling time was 5, 10, 15, 20, and 25 min.

**Table 1 pharmaceutics-14-02173-t001:** Overview of prepared MLX mixtures with SLS. R—reference, CM—co-milled samples.

Grinding Time (min)	SLS Content (wt.%)			
	0	5	15	25
0	R0	-	-	-
5	R10-0/5 R2-0/5	-	CM2-15/5 CM10-15/5	-
10	R10-0/10 R2-0/10	CM10-5/10	CM2-15/10 CM10-15/10 R10-15/10 *	CM10-25/10
15	R10-0/15 R2-0/15	-	CM2-15/15 CM10-15/15	-
20	R10-0/20 R2-0/20	-	CM2-15/20 CM10-15/20	-
25	R10-0/25 R2-0/25	-	CM2-15/25 CM10-15/25	-
30	R2-0/30	-	CM2-15/30	-
35	-	-	CM2-15/35	-

* Reference sample prepared by mixing micronized MLX (R10-0/10) with SLS.

**Table 2 pharmaceutics-14-02173-t002:** Sauter diameter (*D*) [3,2] and specific surface area (*A*_s_) obtained by laser diffraction for the MLX reference samples milled under different conditions with two sizes of grinding balls (2 and 10 mm) and two mill rotation speeds (100 and 300 rpm).

Grinding Time (min)	*D* [3,2] (μm)			*A*_s_ (m^2^/kg)		
	2 mm	10 mm		2 mm	10 mm	
	100 rpm	100 rpm	300 rpm	100 rpm	100 rpm	300 rpm
0 *	5.58 ± 0.06	5.58 ± 0.06	5.58 ± 0.06	666 ± 7	666 ± 7	666 ± 7
5	4.51 ± 0.05	3.81 ± 0.06	4.45 ± 0.81	825 ± 9	975 ± 15	836 ± 153
10	4.06 ± 0.08	3.77 ± 0.12	3.75 ± 0.23	917 ± 17	986 ± 30	992 ± 61
15	3.69 ± 0.10	3.37 ± 0.08	3.25 ± 0.16	1009 ± 28	1104 ± 24	1143 ± 54
20	3.32 ± 0.04	3.20 ± 0.05	-	1122 ± 12	1162 ± 17	-
25	3.23 ± 0.02	3.09 ± 0.01	-	1150 ± 8	1205 ± 2	-
30	3.09 ± 0.02	-	-	1203 ± 6	-	-

* Untreated MLX sample (R0).

**Table 3 pharmaceutics-14-02173-t003:** Values of *k*, *b*, and *T*_0_ parameters calculated according to the Weibull model for MLX samples.

Sample	*k* (min^−b^)	*b* (−)	*T*_0_ (min)
R0	0.0027 ± 0.0011	0.63 ± 0.05	0.3 ± 0.1
R10-0/5	0.0019 ± 0.0007	0.93 ± 0.07	~0
R10-0/10	0.0022 ± 0.0004	0.90 ± 0.04	0.1 ± 0.1
R10-0/15	0.0060 ± 0.0003	0.73 ± 0.09	0.3 ± 0.1
R10-15/10 *	0.17 ± 0.03	0.62 ± 0.07	0.2 ± 0.1
CM10-5/10	0.22 ± 0.02	0.70 ± 0.11	0.3 ± 0.1
CM10-15/10	0.29 ± 0.05	0.92 ± 0.02	~0
CM10-25/10	0.31 ± 0.06	0.75 ± 0.02	0.1 ± 0.1

* Reference sample prepared by mixing micronized MLX (R10-0/10) with SLS.

**Table 4 pharmaceutics-14-02173-t004:** Dependence of the values of parameters *k*, *b*, and *T*_0_ calculated according to the Weibull model for co-milled MLX samples with 15 wt.% SLS with two sizes of grinding balls (2 and 10 mm) on the co-milling time.

Co-Milling Time (min)	*k* (min^−b^)		*b* (−)		*T*_0_ (min)	
	2 mm	10 mm	2 mm	10 mm	2 mm	10 mm
5	0.21 ± 0.07	0.22 ± 0.01	0.61 ± 0.06	0.63 ± 0.01	0.6 ± 0.1	0.1 ± 0.1
10	0.19 ± 0.01	0.26 ± 0.01	0.69 ± 0.01	0.67 ± 0.01	0.6 ± 0.1	0.1 ± 0.1
15	0.21 ± 0.01	0.32 ± 0.01	0.66 ± 0.01	0.72 ± 0.01	0.6 ± 0.1	~0
20	0.23 ± 0.03	0.29 ± 0.01	0.90 ± 0.02	0.70 ± 0.01	~0	0.3 ± 0.1
25	0.24 ± 0.01	0.30 ± 0.02	0.84 ± 0.06	0.73 ± 0.02	0.1 ± 0.1	0.1 ± 0.1
30	0.33 ± 0.07	-	0.79 ± 0.08	-	~0	-
35	0.21 ± 0.01	-	0.82 ± 0.08	-	0.1 ± 0.1	-

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
