# Peer review of "Dissolution Kinetics of Meloxicam Formulations Co-Milled with Sodium Lauryl Sulfate"

_pharmaceutics, 2022, doi:10.3390/pharmaceutics14102173_

Round 1

Reviewer 1 Report

The manuscript “Dissolution kinetics of Meloxicam formulations co-milled with sodium lauryl sulfate” is an interesting piece of work that fits very well in the "Recent Progress in Formulation Approaches for Improving the Solubility and Bioavailability of Poorly Soluble Drugs" Special Issue of Pharmaceutics.

The Authors have performed a substantial amount of experiments combining them with relevant computations. The methods are clearly presented enabling other researchers to reproduce the results and reuse the methodology for other cases. The results are thoroughly described and the studied phenomena are tackled with a variety of experimental and theoretical approaches. The conclusions are fully supported by the results and while they are focused on a single specific active pharmaceutical ingredient they offer some valuable general considerations. Overall, the scientific merit of this study is high, the topic is interesting and the presentation is well executed.

I have however one question regarding the presentation of the results.  The Authors indicate that using 2 mm grinding balls is the optimal choice as they provide more surface activation than 10 mm balls.  Meanwhile, in Figure 7 we can see SEM images obtained when using 10 mm balls. Should a difference be expected when 2 mm balls would be used? Also, in Figure 8 the images are taken for the SLS content equal 5 wt. %, while the optimal composition was found to be 15 wt. %. Again, should a difference be expected? It seems natural to me to present such images taken under optimal conditions so please clarify this small issue. Furthermore, image 7b is not described in the text.

Also, there are some minor grammatical and editorial issues, please check the whole manuscript.

Line 64: “(“ instead of “/”

Line 221: space after „100”

Line 261: „which" instead of „whose”

Line 258: „mixtures” as in plural

Lines 402-403: either „the formation of MLX aggregates occurs” or “MLX aggregates are formed”

Lines 458 and 459: space after “10”

Figure 10 is not properly embedded in the file

Author Response

Answers to Comments of Reviewer #1:

The Authors indicate that using 2 mm grinding balls is the optimal choice as they provide more surface activation than 10 mm balls.  Meanwhile, in Figure 7 we can see SEM images obtained when using 10 mm balls. Should a difference be expected when 2 mm balls would be used?

The use of these SEM images was chosen because Figure 7 follows the previously discussed results and graph (Fig. 6). I suppose that images for 2 mm balls would be similar to Fig. 7c based on other results. Probably there would not be such differences comparing SEM images of only samples milled with 2mm balls. The goal here was to show the significant difference between individual samples from the point of view of particle dispersion and the formation of composites.

Also, in Figure 8 the images are taken for the SLS content equal 5 wt. %, while the optimal composition was found to be 15 wt. %. Again, should a difference be expected? It seems natural to me to present such images taken under optimal conditions so please clarify this small issue.

Thanks for pointing out the error. The image was mislabeled. For elemental EDS analysis, a sample with optimal properties, i.e. CM10-15/10, was selected.

Furthermore, image 7b is not described in the text.

Added to the text.

Also, there are some minor grammatical and editorial issues, please check the whole manuscript.

Line 64: “(“ instead of “/”

Line 221: space after „100”

Line 261: „which" instead of „whose”

Line 258: „mixtures” as in plural

Lines 402-403: either „the formation of MLX aggregates occurs” or “MLX aggregates are formed”

Lines 458 and 459: space after “10”

Figure 10 is not properly embedded in the file

Thank you for checking the document. We have corrected all the suggested issues and again checked the manuscript for grammatical and editorial issues.

Reviewer 2 Report

The manuscript's authors investigate the effect of milling and co-milling technique using SLS to enhance The dissolution of Meloxicam (MLX),  which  a poorly soluble drug to facilitate the

manufacture of rapidly releasing dosage forms. The study is focused on investigating the dissolution kinetics of particulate formulations, used XRD data to detect the crystalline nature of MLX . The shape of particles also detected by SEM . The figure and the table are informative, but some correction are necessary before publication. Several research articles discussed this idea before. 

Author Response

Answers to Comments of Reviewer #2:

  1. Abstract : line 9. The expression challenging is not suitable owing to, there is huge research about meloxicam as a rapid release dosage forms as DOI: 10.3109/03639045.2014.922573 J. KOr, pharm. sci. Vol.35, No.3 (2005?

We thank to the reviewer for pointing out that our statement may be understood in a way that the acceleration of the dissolution of meloxicam was not studied at all in the literature, which is misleading. We focus on co-milling specifically in our introduction, but the reader should be informed of other alternative approaches. Therefore, the abstract was edited, and the recommended reference was added to the introduction section of the manuscript.

  1. Line 12. ( 5-25 wt %) of SLS

Corrected.

  1. Introduction: line 87 it is not the only study there are others like
  • 1- Meloxicam-PVP-SLS ternary dispersion systems: In-vitro and in-vivo evaluation.

January 2010International Journal of Pharmacy and Pharmaceutical Sciences

Vol.2(1):[Page 182-190

  • INFORMATION FOR THE USER (Meloxicam 15 mg Orodispersible Tablets,

Meloxicam 15.0 mg ( contain SLS) )

Thanks for the reminder. A reference to the first study was added to the introduction part of our manuscript, and a  discussion and comparison of our results with previously published data from other authors have been added to the text.

  1. Line 97 materials and methods ; Preferred to write SLS as all words in an article.

It was corrected in the whole manuscript.

  1. Line 111: Particle size analysis: correct tween to tween 80.

Corrected.

  1. Line 119 (Micronization and co-milling of Meloxicam ) : How much mg of MLX used, It must be mentioned.

The weight of MLX for milling experiments was added.

  1. Table 1 line 148 : The symbols in the table need more explanation.

Thanks for the notice. I agree that simplifying the description may have been misleading. In correction of the text, I tried to provide a more detailed description of the labeling of our samples.

  1. Dissolution testing, line 164 : how much mg of MLX.

A detail of the weight for dissolution measurements was added.

  1. and 10. Result and discussion : line 185, Dissolution testing .It is preferred to unite the word Meloxicam or MLX in all article and no (an ) for SLS grammar chick.

Line 187: SLS preferred the author write sodium lauryl sulphate and other time SLS (unite the word)

Thank you for the notices. We unified the labeling of substances in the entire manuscript.

  1. Line 190 : must be indicated by DSC besides X ray to indicate (amorphization of the drug).

Since our goal was to compare dissolutions in a wide range of the entire sample, the goal was not to quantify the content of the amorphous portion within the surface changes. Physico-chemical characterization and especially XRD analysis was intended from the entire volume so as to show whether the material is similar from a crystallographic point of view. However, we agree that it would be interesting to perform additional measurements, such as DSC or surface energy analysis, which would tell us more about minor but important changes. Since the project had already ended, further measurements were difficult to provide with in such a short time, and only information better describing the XRD measurements was added to the article.

  1. Line 223; From the table, there is a reduction in size with time and rpm the authors must complete the results of 300 rpm at 20, 25, and 30 min and indicate the word ( does not have such an influence ) by significance or non significance and at what P.

One of the main comparisons was made for samples milled for 10 minutes, therefore, milling experiments with higher rpm were not performed. As already mentioned, the project has ended, and further measurements were difficult to provide with in such a short time. The comparison was made only on the basis of the obtained results, which are presented in Table 2, and where the values of the Sauter diameter after 10 minutes hardly differ.

13 Line 250: Is it enough to calculate the mean ?.

Information on the number of repetitions of dissolution measurements was added to the text. We believe that 3 measurements are sufficient to get an overview of the improvement of the dissolution rate between the co-milled samples.

  1. Line 257: Fig 2: fig. must include pure MLX without any treatment.

Thank you for the notice. Figure 2 was changed, and the dissolution curve for a pure MLX was added.

  1. - Line 262 : Why use this model only, there are other seven models such as zero order, first order, Higuchi, Hixson-Crowell, the square root of mass, and the three-second root of mass, the authors can compare the results for these eight models but only on model is not enough.

We agree with the reviewer that the mentioned models are suitable for comparing different types of dissolution experiments. In our case, the goal was not to compare different models but to interpolate the experimental points as best as possible and obtain the parameters that we would discuss. In this case, the Weibull model proved to be the most appropriate.

  1. Line 310: are the authors calculated the statistic difference , and at what p value.

Unfortunately, a deeper static analysis of the data was not part of this study.

  1. Line 318 : the smallest value is R10-15/10 of b nor R0.

Thanks for pointing this out. I agree that the description of the values was confusing, and the text was corrected.

  1. Line 327: where is R0 in figures, ad pure without micronization must be added, the result mentioned is (R10 - 0/10) and the significance at what P value.

According to reviewer #3, the values of parameters in Table 3 have been recalculated, and standard deviations were added. Dissolution curve of the pure MLX without micronization was added to the Figure 2.

  1. Fig 3 ; Is packed with curves, these curves need to be revised and clarified.

Both images have been altered to show the differences between individual samples better.

  1. Fig. 4 : the figure is not clear, They are very confusing, these curves need to be revised and simplified.

Figure 4 was changed, and a more detailed description was added to the text. Thank you for the notice.

Reviewer 3 Report

Review comments on pharmaceutics-1918648: Dissolution kinetics of Meloxicam formulations co-milled with sodium lauryl sulfate

The manuscript described the preparation and characterization of meloxicam co-milled with SLS. The authors prepared different formulations with varying SLS content, milling ball size, and milling time. Dissolution data were fitted to the Weibull model to evaluate the release kinetics. Overall, the study was appropriately designed and implemented. However, there are several issues with methods and data presentation. The detailed comments are as follows.

1. Dissolution test: was the sink condition achieved in this study? Why was pH 7.2 used instead of pH 1.2 (gastric pH)?

2. UV-VIS spectroscopy: did the SLS affect MLX absorption at 271 nm and 363 nm? Did the authors evaluate the specificity, accuracy, and precision of the method?

3. Several studies on MLX co-milling are available, as mentioned in the Introduction. The authors should discuss the findings in this study and previous ones. Is SLS an excellent option to improve MLX dissolution compared with other co-former?

4. Section 3.2.1, the part in lines 255-292 should be moved to the Method section. Section 3.2.2, the part in lines 380-389 should be moved to the Method section and reference should be added. Section 3.2.3, the part in lines 408-444 should be moved to the Method section.

5. Experiment/ measurement should be repeated with n≥3. For example, data in Table 2 should be Means of ≥3 measurements. The authors should mention the number of replications in the dissolution test.

6. Tables 3 and 4: The authors should present R^2 of the data fitting to the Weibull model. K, b, and T0 values should be means±SDs with n = the number of replications in the dissolution test.

7. Figure 5: The authors should calculate separate r values and present data as means±SDs with n = the number of replications in the dissolution test.

8. There was no mention of XRD in the Method section.

8. Figures 5, 6, 9, and 10: The authors should correct the decimal errors in Y-axis.

9. Figure 10 is of low quality.

Author Response

Answers to Comments of Reviewer #3:

  1. Dissolution test: was the sink condition achieved in this study? Why was pH 7.2 used instead of pH 1.2 (gastric pH)?

I agree with the reviewer that for instant release, it would be more appropriate to use gastric pH. We have designed our experiments to be comparable to other similar works in the literature, where only neutral pH appears almost exclusively, as recommended by the FDA for testing formulations with Meloxicam.

  1. UV-VIS spectroscopy: did the SLS affect MLX absorption at 271 nm and 363 nm? Did the authors evaluate the specificity, accuracy, and precision of the method?

Since SLS almost does not absorb in the UV region, under the conditions we used in the concentration range of SLS in the mixtures, it did not affect the analyses.

  1. Several studies on MLX co-milling are available, as mentioned in the Introduction. The authors should discuss the findings in this study and previous ones. Is SLS an excellent option to improve MLX dissolution compared with other co-former?

Thanks for the reminder. A discussion and comparison of our results with previously published data from other authors have been added to the text.

  1. Section 3.2.1, the part in lines 255-292 should be moved to the Method section. Section 3.2.2, the part in lines 380-389 should be moved to the Method section and reference should be added. Section 3.2.3, the part in lines 408-444 should be moved to the Method section.

We agree with the reviewer that these paragraphs could be part of the method section. We decided to keep them in the text for greater clarity and the gradual construction of the discussion of our results.

  1. Experiment/ measurement should be repeated with n≥3. For example, data in Table 2 should be Means of ≥3 measurements. The authors should mention the number of replications in the dissolution test.

Information about the number of experiment repetions was added to the text. There were always at least 3 repeated measurements for the co-milled samples, and the results are presented as the mean of all measurements.

  1. Tables 3 and 4: The authors should present R^2 of the data fitting to the Weibull model. K, b, and T0 values should be means±SDs with n = the number of replications in the dissolution test.

I agree with the reviewer that it was not appropriate to calculate the model from the mean values of concentration obtained from the dissolution experiments. According to the recommendation, each measurement was interpolated with a model, and the mean values with standard deviations from the individual obtained parameters of the Weibull model were presented in the tables.

  1. Figure 5: The authors should calculate separate r values and present data as means±SDs with n = the number of replications in the dissolution test.

Figure 5 does not present new data, but it is a different view of the data from the earlier figures. As can be seen from Equation 2, the r value is calculated from one statistical variable, which is statistically processed in Figures 2 and 3, including standard deviations; and two constants. The principle is to visualize another aspect and show what is harder to see elsewhere. Since the data in Fig. 5 differ significantly in values, and a break on y-axis is used, we think it would be confusing to present the error bars.

  1. There was no mention of XRD in the Method section.

Details of XRD measurements were added.

  1. Figures 5, 6, 9, and 10: The authors should correct the decimal errors in Y-axis.

Thank you for the notice. All the graphs were corrected and uploaded in the new version of the manuscript.

  1. Figure 10 is of low quality.

Figure 10 was changed. Thank you for the notice.

Round 2

Reviewer 2 Report

Thanks for the effort done for the major correction, but minor edits needed

Author Response

Answers to Comments of Reviewer #2:

  1. Why DSC study not included the author not mention any comment.

We completely agree with the reviewer’s comment that the DSC measurements may be more sensitive to detect possible partial amorphization. However, that was not an objective of our work as our goal was to compare dissolutions in a wide range of the entire sample, and not to quantify the content of the amorphous portion within the surface changes. Physico-chemical characterization and especially XRD analysis was intended from the entire volume so as to show whether the material is similar from a crystallographic point of view. So we tried to add this explanatory information to the article.

However, we agree that it would be interesting to perform additional measurements, such as DSC or surface energy analysis, which would tell us more about minor but important changes and we thank to the reviever for suggesting this option as we can perform the measurement in a follow up study. Since the project had already ended, further measurements would difficult to provide with in such a short time. Their addition would also split the focus of the paper and opened additional discussion, which will be better to support also by surface energy measurement and discuss in another article. Thus, the addition of DSC data is not possible at this time and, only the information better describing the XRD measurements was added to the article.

  1. The values of P for significance not mentioned an in all paper (must be mentioned )

Thank you for the notice. Wherever appropriate in the whole article, we made comparisons of the differences in measured values between pairs on the basis of a t-test of mean values and standard deviations. The p-values were added to the discussion of the results.

  1. The conclusion must be concise

We thank to the reviewer for pointing out that the conclusions were indeed too extensive. We tried to condense the conlusion by moving paragraphs which summarized specific parts of the discusion into the Results and Discussion section.

Reviewer 3 Report

Review comments on pharmaceutics-1918648: Dissolution kinetics of Meloxicam formulations co-milled with sodium lauryl sulfate

The manuscript was revised and partly improved. However, there are still several issues to consider as follows (they are issues in the previous review report that the authors ignored).

1. Dissolution test: was the sink condition achieved in this study?

2. UV-VIS spectroscopy: the authors should add evaluation data of the specificity, accuracy, and precision of the method to ensure the reliability of the measurements.

3. Section 3.2.2: the part in lines 416-425 needs references.

4. Table 2: were data values of single measurements? The authors should present Means ± SDs of ≥3 measurements.

Author Response

1) Dissolution test: was the sink condition achieved in this study?

First of all, I would like to apologize for forgetting to respond to some questions or comments. Thank you for the good question. Yes, the sink condition was achieved. Each dissolution experiment was conducted using the flow-through cell in an open-loop mode with a fresh dissolution medium from the reservoir continuously passing through the cell. These settings are in the literature described as infinite sink conditions.

(Zongming G., In Vitro Dissolution Testing with Flow-Through Method: A Technical Note, AAPS PharmSciTech, 2009, 10(4): 1401)

2) UV-VIS spectroscopy: the authors should add evaluation data of the specificity, accuracy, and precision of the method to ensure the reliability of the measurements.

To determine the content of MLX in the samples taken from the dissolution experiments, 10 calibration solutions of MLX were prepared in the dissolution medium, so that the samples taken were always within the calibration concentration range. Calibration lines were constructed based on the entire absorption spectrum, where MLX showed two maxima at 271 and 363 nm wavelengths, corresponding to published literature data. The coefficient of determination of both calibration lines was higher than 0.9995. Before each experiment, it was verified that the dissolution medium itself does not affect the measurement, i.e., its absorbance at the monitored wavelengths (271 and 363 nm) was in the range of 0.0005+-0.0003 (intercept on the calibration line), which corresponds to less than 0.5 wt.% of the studied sample. Furthermore, the addition of SLS to the calibration standards was monitored. Since SLS hardly absorbs at all at the given wavelengths, the change in absorbance was almost unmeasurable (0.0005).

3) Section 3.2.2: the part in lines 416-425 needs references.

Thank you for the notice. We added a reference.

4) Table 2: were data values of single measurements? The authors should present Means ± SDs of ≥3 measurements.

I would like to apologize that this table was not already modified in the first correction. Table 2 has therefore been revised, and shows mean values and SDs calculated from 15 values. As described in section 2.2, the determination of PSD and Sauter diameter was performed in three series with five measurements of each sample.
